# Effect of osteoporosis-related reduction in the mechanical properties of bone on the acetabular fracture during a sideways fall: A parametric finite element approach

**Shahab Khakpour** [1]* , **Amir Esrafilian** [2], **Petri Tanska** [2‡], **Mika E. Mononen** [2‡], **Rami K. Korhonen** [2‡], **Timo Jämsä** [1,3,4]

**1** Research Unit of Medical Imaging, Physics, and Technology, University of Oulu, Oulu, Finland, **2** Department of Applied Physics, University of Eastern Finland, Kuopio, Finland, **3** Medical Research Center, University of Oulu and Oulu University Hospital, Oulu, Finland, **4** Diagnostic Radiology, Oulu University Hospital, Oulu, Finland

☯ These authors contributed equally to this work.
‡ PT, MEM and RKK also contributed equally to this work.
* Shahab.khakpour@oulu.fi

**Data Availability Statement:** All raw data, including the geometry, material etc. and samples of simulation input files to replicate the simulation

## Abstract

### Purpose

The incidence of acetabular fractures due to low-energy falls is increasing among the geriatric population. Studies have shown that several biomechanical factors such as body configuration, impact velocity, and trochanteric soft-tissue thickness contribute to the severity and type of acetabular fracture. The effect of reduction in apparent density and elastic modulus of bone as well as other bone mechanical properties due to osteoporosis on low-energy acetabular fractures has not been investigated.

### Methods

The current comprehensive finite element study aimed to study the effect of reduction in bone mechanical properties (trabecular, cortical, and trabecular + cortical) on the risk and type of acetabular fracture. Also, the effect of reduction in the mechanical properties of bone on the load-transferring mechanism within the pelvic girdle was examined.

### Results

We observed that while the reduction in the mechanical properties of trabecular bone considerably affects the severity and area of trabecular bone failure, reduction in mechanical properties of cortical bone moderately influences both cortical and trabecular bone failure. The results also indicated that by reducing bone mechanical properties, the type of acetabular fracture turns from elementary to associated, which requires a more extensive intervention and rehabilitation period. Finally, we observed that the cortical bone plays a substantial role in load transfer, and by increasing reduction in the mechanical properties of cortical bone, a greater share of load is transmitted toward the pubic symphysis.

were added as Data sharing links to this submission. However, the clinical CT images cannot be shared publicly due to data sensitivity according to the research permit granted by the Northern Ostrobothnia Hospital District, Oulu, Finland. The possibilities for data sharing outside the project team will be evaluated case by case. The contact person for register-based data in Northern Ostrobothnia Hospital District is Miia Turpeinen (miia.turpeinen@ppshp.fi), Senior Medical Officer, Head of Unit, Research services, Oulu University Hospital.

Funding: Shahab Khakpour: This project received funding from European Union's Horizon 2020 research and innovation programme under the Marie Skoldowska-Curie grant agreement No 713606. https://www.oulu.fi/i4future/ The funders had no role in study design, data collection and analysis, decision to publish, or preparation of the manuscript.

Competing interests: The authors have declared that no competing interests exist.

## Conclusion

This study increases our understanding of the effect of osteoporosis progression on the incidence of low-energy acetabular fractures. The osteoporosis-related reduction in the mechanical properties of cortical bone appears to affect both the cortical and trabecular bones. Also, during the extreme reduction in the mechanical properties of bone, the acetabular fracture type will be more complicated. Finally, during the final stages of osteoporosis (high reduction in mechanical properties of bone) a smaller share of impact load is transferred by impact-side hemipelvis to the sacrum, therefore, an osteoporotic pelvis might mitigate the risk of sacral fracture.

## 1. Introduction

Osteoporosis (OP) is the primary cause of 1.5 million fractures per year [1]. Up to 50% of women and 33% of men experience an osteoporotic fracture in their life [2]. OP-related morbidities and mortalities impose a heavy economic burden on societies and healthcare systems annually and affect the life quality of patients and their families [3]. For instance, by 2025, the direct costs of OP-related complications in the United States are expected to increase to $25.6 billion per year [4].

The incidence of low-energy acetabular fractures has increased 2.4 times in developed countries such as the United States, Finland, and Sweden in recent decades [5–8]. While pelvic fractures among youngsters and adults typically result from high-energy traumas such as motor vehicle accidents [9], low-energy acetabular fractures are mostly observed among elderly people. Most low-energy acetabular fractures occur because of falls from a standing height [9], which rarely cause a fracture in young persons but can be detrimental for elderly people. Owing to the lower incidence of low-energy acetabular fractures in comparison with low-energy proximal femur fractures [10], the former has been rarely studied, and there is a research gap in the biomechanics of low-energy acetabular fractures. The main issue in studying low-energy fracture mechanisms is related to the lack of experimental data since the fractures typically occur during daily activities, and those conditions may be difficult or unethical to replicate *in vivo*. An alternative solution for this is the evaluation of fracture mechanisms through computational simulations that incorporate typical conditions and forces during falling. By including patient-specific data such as bone density distribution derived from computed tomography (CT), computational models can assess potential risks for bone fractures during falling [11–13]. One such computational method is finite element modeling and analysis, which can address the questions related to relationships between applied forces and mechanical responses of tissue (stresses and strains). To the best of our knowledge, the studies conducted by Shim et al. [14] and our group [15, 16] are the only finite-element studies focusing on low-energy acetabular fractures at the tissue level. Our previous studies revealed that the effects of impact velocity and body configuration at the time of impact may substantially contribute to the severity and type of acetabular fractures. Also, the trochanteric soft-tissue thickness was suggested to be more important in the prevention of low-energy acetabular fractures than trochanteric soft-tissue stiffness or flooring material type [15, 16]. In these studies, the bone was assumed to be healthy, and the effect of reduction in the mechanical properties of bone caused by OP on the severity and type of low-energy acetabular fracture was not investigated.

The strength of the cortical and trabecular bones, as the primary structures of the bone, is affected by material composition, organization, and the resulting anisotropic material properties [17]. OP significantly changes the bone structure and is seen as the loss of bone mass (mineral content) and reduction in the bone volume fraction [18]. During OP progression, the cortical bone becomes thinner and more porous [17], and the strut and plate structures of the

trabecular bone turn into rod-like elements with increased void spaces [19]. While the main constituents of the cortical and trabecular bones are similar, under an equivalent bone remodeling rate, the trabecular bone may lose more bone mass due to its greater surface to volume ratio, although this trend could be changed by aging and intracortical porosity level [20, 21]. Whereas OP mainly causes trabecular bone fractures in patients below age 65, older patients who may have lost a considerable part of their trabecular bone, are at a higher risk of cortical bone fractures [22]. The anatomical location of the bone plays a substantial role in how the cortical and trabecular bones contribute to the bulk load-bearing properties of the bone. For instance, the contribution of the cortical bone at regions experiencing bending loads and the role of the trabecular bone at the areas under axial loading are critical [17]. The pelvis is an irregular bone with a sandwich-like structure (a thin cortical bone enveloping the trabecular bone), and it is subjected to complex loading [23]. These changes are reflected in the reduction in the mechanical properties (e.g. elastic modulus) of cortical and trabecular bone [24, 25].

In addition to pharmacologic means (anabolic and antiresorptive therapy) [26], preventive measures are critical in the reduction of osteoporotic fractures incidence [3]. The number of osteoporotic fractures can be remarkably reduced through early actions such as regular screening, muscle-strengthening exercises, and anabolic therapy [27]. However, these preventive methods should be designed to protect the trabecular or cortical bones according to their anatomical location and the age of the patient [17]. Thus, it would be crucial to characterize the effect of reduction in the elastic moduli of trabecular and cortical bone on low-energy acetabular fractures.

This study aimed to assess the contribution of reduction in the mechanical properties of trabecular and cortical bone caused by OP on the acetabular bone failure and load-transfer mechanisms within the pelvic ring. Toward this goal, as a reliable approach in bone fracture prediction [11, 15, 16, 28, 29], a series of parametric finite element simulations of reduction in the mechanical properties of trabecular, cortical, and total bone (trabecular and cortical simultaneously) was done. The results of this study increase our knowledge about the effect of OP-related reduction in the mechanical properties of bone on the risk of acetabular fractures and load-transfer mechanisms within the pelvic ring.

## 2. Materials and methods

This study is based on a 3D model of the human femur and pelvis derived from a database [6]. The model preparation and validation steps and the used materials were comprehensively presented earlier [16] and explained briefly here. This study was granted a register-based study permit (No. 220/2017) from the Northern Ostrobothnia Hospital District, Oulu, Finland.

### 2.1. 3D model preparation

The model was reconstructed from the abdominal computational tomography (CT) of a 50th-percentile of anthropometric data (such as pubic arch angle, pubic ramus width, and pubic symphysis length) from male patients (without any hip or pelvic fracture history) derived from a larger study database [6]. The database was gathered by the Oulu University Hospital from non-fractured patients who came to the clinical abdominal CT imaging (without using a calibration phantom). The cortical, trabecular, and trochanteric soft tissues of the pelvic ring were thresholded and segmented using Mimics® (version 21.0, Materialise Software, Belgium). The method assuring precise segmentation of cortical bone, especially in low-thickness regions such as acetabulum, was explained in our previous work [16]. Similar to the approach used by Majumder et al. [12], the highest and lowest Hounsfield Units (HU) were assumed to be corresponding to the apparent density of cortical bone (1.8 g/cm$^3$) and marrow cavity (0.01 g/cm$^3$). Element by element density assignment for trabecular bone was done by using the relationship presented by

Rho et al. [30] between HU and apparent density and material-assignment feature of Mimics®. In the absence of a contrast agent during imaging, based on anatomical (Human Biodigital® online platform [31]) and the published data, the femoral head [32], acetabular [32], and sacroiliac cartilages [33], as well as the interpubic disc [34], were built by using 3-matic® (version 13.0, Materialise Software, Belgium). The trochanteric soft tissue covering the bony parts was segmented and reconstructed directly from the CT. The nonimpact side of the trochanteric soft tissue was excluded to reduce computational costs [12, 35] (Fig 1A).

Our previous studies revealed that body configuration affects strain magnitude and distribution within the acetabulum substantially, and based on that, the body configuration resembling the highest risk of acetabular fracture was considered for this study [15] (Fig 1A). Since the imaging was done in the supine position, the bony parts of the model and the attached trochanteric soft tissue were rotated to achieve a sideways fall configuration [16]. Also, the upper and lower extremities were modeled as a lumped mass-spring-dashpot system [16].

Since the incidence of isolated acetabular fractures after a fall on their side is significantly higher among elderly people than in other age groups and the concomitant nonacetabular injuries are mostly resulted from a high-energy trauma [36], only the impact-side hemipelvis was studied here. Also, to increase the accuracy at the region of interest and reduce overall computational costs, only the bony parts at the impact-side hemipelvis meshed with quadratic 10-node tetrahedral solid elements, and the other parts meshed with linear 4-node tetrahedral solid elements [11]. The converged mesh [16] consists of 3,182,326 solid elements (element edge size ranging between 1.16 and 3.47 mm).

## 2.2. Mechanical properties of the healthy bone and other tissues

Cortical and trabecular bones were modeled as strain-rate sensitive Fu-Change Foam based on the method and empirical relationships proposed by Enns-Bray et al. [37], available in Appendix A in S1 File. Also, examples of the stress-strain curves achieved based on this method and used as the material model input in the finite element model are presented in Appendix A in S1 File. Both cortical and trabecular bones were modeled as viscoelastoplastic (strain-rate dependent with strain-rate coverage range: 0.008–30 s$^{-1}$) materials with different behaviors in tension and compression [37]. Since the cortical bone has rather uniform mechanical properties [17], it was considered as a homogeneous material, while the trabecular bone was assumed to be heterogenic (Appendix A in S1 File) and was implemented into the model using the material mapping strategy available in Mimics® [12]. By using the empirical relationship between apparent density (derived from the CT) and elastic modulus ($E = 6850 \times \rho_{app}^{1.49}$ (MPa)) proposed by Morgan et al. [38], the elastic modulus was calculated for each element. Also, by knowing the apparent densities and employing the empirical relationship used by Enns-Bray et al. [37], the remaining mechanical properties, such as proportionality limit, yield, and ultimate stresses/strains, were calculated for the trabecular and cortical bones (Appendix A, Table A1 in S1 File).

The articular and sacroiliac cartilages, as well as trochanteric soft tissue, were modeled as hyperelastic materials. The ground was assumed to be rigid (Appendix A in S1 File). According to our previous study [16] and similar to Fleps et al. and Majumder et al. [11, 12, 29], the viscous damping of trochanteric soft tissue during a lateral impact can be ignored. Therefore, the experimentally validated material model proposed by Majumder et al. [12] for simulation of sideways falls on the greater trochanter was used in this study.

## 2.3. Mechanical properties of the osteoporotic bone

For simulating the effect of reductions in the mechanical properties of bone, we applied the isotropic osteoporosis (IO) model [25, 39], with the elastic moduli reductions of the cortical

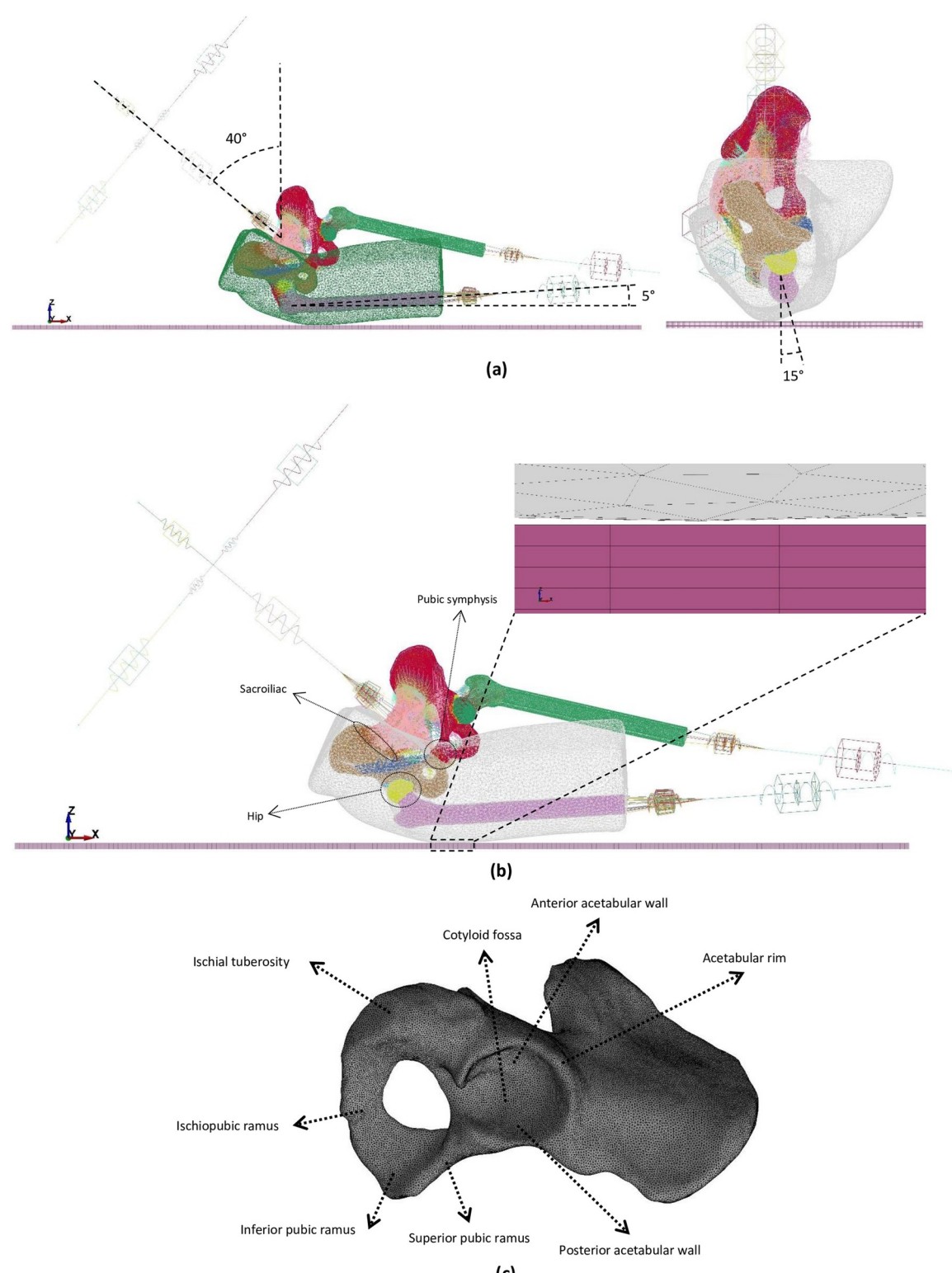

**Fig 1.** a) Sideways fall configuration. b) The location of studied joints and the initial position of the model. c) Pelvic anatomical landmarks.

and trabecular bones by 33% and 66%, respectively. IO model was proposed by Polikeit et al. [39] based on the difference between bone mineral density and elastic modulus of healthy and osteoporotic bones. The model was based on previous literature, such as Dickenson et al. [40] who mechanically tested the cortical bone strength, and Mazess [41] who reported the trabecular bone loss rate during aging.

In the present study, relative reductions in elastic moduli of the trabecular ($E_{Trab-OP}/E_{Trab-Normal}$), cortical ($E_{Cort-OP}/E_{Cortic-Normal}$), and total bone ($E_{Total-OP}/E_{Total-Normal}$) caused by OP were simulated (Table 1). According to the IO model, the cases with the ultimate reduction in the trabecular and cortical elastic modulus were defined as $E_{Trab-OP}/E_{Trab-Normal} = 0.34$ and $E_{Cort-OP}/E_{Cortic-Normal} = 0.67$ respectively. In each simulation, the corresponding reduced apparent density (bone loss) was calculated first, and then by using the experimental relationship proposed by Enns-Bray et al. [37] (available in Appendix A in S1 File), the reduced mechanical properties of bone such as ultimate stress/strain were achieved accordingly. To evaluate the level of contribution of trabecular and cortical bone on the acetabular bone failure and the load-transferring mechanism within the pelvic ring, first, while the cortical bone was assumed to remain intact ($E_{Cort-OP}/E_{Cortic-Normal} = 1$), $E_{Trab-OP}/E_{Trab-Normal}$ was reduced from 0.67 to 0.34, and at the second step, by considering healthy trabecular bone ($E_{Trab-OP}/E_{Trab-Normal} = 1$), $E_{Cort-OP}/E_{Cort-Normal}$ was reduced from 0.835 to 0.67 (Table 1). Finally, by reducing both cortical and trabecular elastic modulus simultaneously, the effect of reduction in elastic modulus of total bone due to OP was simulated.

## 2.4. Initial and boundary conditions

The model was brought to the vicinity of the ground (0.1 mm) to reduce the simulation time (Fig 1B). The impact velocity was set to 3.17 m/s. This velocity was considered as the typical impact velocity resulting from a low-energy sideways fall in similar experimental and computational investigations [11, 12, 35] and introduced as the critical impact velocity in our previous work [16]. Also, the gravitational acceleration (9.806 m/s$^2$) was applied to all moving parts.

**Table 1. Simulated cases with different types and levels of reduction in the elastic modulus.**

| Simulation No. (affected bone) | $E_{Trab-OP}/E_{Trab-Normal}$ | $E_{Cort-OP}/E_{Cort-Normal}$ |
|---|---|---|
| 1 (Trab.) | 0.670 | 1 |
| 2 (Trab.) | 0.614 | 1 |
| 3 (Trab.) | 0.538 | 1 |
| 4 (Trab.) | 0.472 | 1 |
| 5 (Trab.) | 0.406 | 1 |
| 6 (Trab.) | 0.340 | 1 |
| 7 (Cort.) | 1 | 0.835 |
| 8 (Cort.) | 1 | 0.802 |
| 9 (Cort.) | 1 | 0.769 |
| 10 (Cort.) | 1 | 0.736 |
| 11 (Cort.) | 1 | 0.703 |
| 12 (Cort.) | 1 | 0.670 |
| 13 (Total.) | 0.670 | 0.835 |
| 14 (Total.) | 0.614 | 0.802 |
| 15 (Total.) | 0.538 | 0.769 |
| 16 (Total.) | 0.472 | 0.736 |
| 17 (Total.) | 0.406 | 0.703 |
| 18 (Total.) | 0.340 | 0.670 |

Trab.: trabecular bone, Cort.: cortical bone, Total.: Trab + Cort., OP: osteoporosis, E: elastic modulus.

All nodes in contact at the cortical-trabecular and cortical-articular cartilages and cortical-trochanteric soft-tissue interfaces were tied in all degrees of freedom. While the contact between the femoral head and acetabulum was assumed to be frictionless [11], the static and kinetic coefficients of friction between the trochanteric soft tissue and the ground were set to 0.5 and 0.36, respectively [12, 42, 43]. The internal reaction force (RCFORCE, LS-DYNA) was calculated based on the resultant contact forces for the slave and master sides of the contact interfaces of the studied joints (e.g., acetabular and femoral head cartilages within the hip joint) [44].

## 2.5. Failure criteria and mechanisms

In the absence of cadaveric test data, this study did not aim to precisely predict bone fracture. This ultimate strain criterion has been widely used in bone fracture prediction [11, 12, 35, 37, 45]. Therefore, the bone failure criteria developed by Enns-Bray et al. [37] and used successfully to replicate cadaveric bone fracture experiences by Fleps et al. [11] were used here too. Briefly, for the trabecular bone, while the ultimate stress was rate-dependent, the ultimate strain was assumed to be rate-independent [37] (Appendix A in S1 File), and compressive and tensile bone failures are initiated by the onset of element softening (first principal strains of the Green-St.Venant strain tensor higher than 0.014 and third lower than -0.02), without any element erosion (deletion) to ensure energy conservation [11, 37]. For the cortical bone, the ultimate strain is considered to be rate-dependent. Therefore, the strain rate range in the acetabulum region was calculated for each simulation, and by calculating the corresponding ultimate strains by using the formulas available in Appendix A in S1 File, the lower band (onset) of compressive and tensile failure was chosen for the cortical bone. Since it is also essential to define an upper band, it was assumed to be equal to -0.1 in compression and 0.07 in tension, similar to the trabecular bone (Appendix A in S1 File).

## 2.6. Analysis

Since the peak impact force typically occurs between 5 and 15 ms [46] after impact initiation, the simulation duration was limited to 60 ms, which has also been observed to be sufficient to let impact energy propagate within the impact-side hemipelvis [11, 12]. The largest stable time step was set to $3.36 \times 10^{-5}$ ms in the simulations.

A total of 18 simulations (Table 1) were conducted in LS-Dyna® (LSTC, Livermore, USA) to investigate the effect of reduction in the mechanical properties of trabecular, cortical, and total bone caused by OP on acetabular bone failure severity and type (according to Judet and Letournel's classification [47] (Appendix A, Fig A5 in S1 File)). In this study to describe different types of fracture, the common term "acetabular fracture" was used. According to Judet and Letournel's definition, acetabular fractures are common at the acetabulum region, but not restricted to it only and include the other regions of the pelvis such as ilium and ischium [47].

## 3. Results

The effect of reduction in the mechanical properties of bone on bone failure (ultimate strain) within the acetabulum and load-transfer mechanisms within the pelvic girdle joints are presented in this section. The adopted pelvic anatomical landmarks are depicted in Fig 1C.

## 3.1. Effect of reduction in the mechanical properties of trabecular bone on the acetabular bone failure

By decreasing $E_{\text{Trab-OP}}/E_{\text{Trab-Normal}}$ from 0.670 to 0.340, compressive bone failure spread in both anterior and posterior acetabular walls (Fig 2). A further reduction in $E_{\text{Trab-OP}}/E_{\text{Trab-Normal}}$ led to

compressive bone failure at the inferior pubic ramus. At the lowest levels of $E_{Trab-OP}/E_{Trab-Normal}$ (0.472–0.340), the superior acetabulum rim also experienced compressive bone failure. Also, the tensile bone failure was minor in comparison with the compressive one. Only the scattered areas at the cotyloid fossa, superior pubic, and ischiopubic ramus failed (Fig 2). Changes in $E_{Trab-OP}/E_{Trab-Normal}$ resulted in neither compressive nor tensile bone failure at the cortical level (Fig 3).

According to the Judet and Letournel acetabular fracture classification, the fracture type at $E_{Trab-OP}/E_{Trab-Normal}$ ranging from 0.670 to 0.538 was posterior/anterior wall, whereas, at $E_{Trab-OP}/E_{Trab-Normal}$ ranging from 0.472 to 0.340, the bone failure pattern resembled both-column and anterior with posterior hemi transfer acetabular fracture types.

### 3.2. Effect of reduction in the mechanical properties of cortical bone on the acetabular bone failure

Compressive trabecular bone failure was observed in all reductions in $E_{Cort-OP}/E_{Cort-Normal}$ levels (0.835–0.670) at the anterior and posterior acetabular walls (Fig 4). The failure region grew slightly by decreasing $E_{Cort-OP}/E_{Cort-Normal}$ (Fig 4). Also, scattered tensile bone failure was observed at the cotyloid fossa and superior pubic ramus (Fig 4). The ultimate reductions in $E_{Cort-OP}/E_{Cort-Normal}$ (0.703–0.670) caused compressive cortical bone failure at the anterior acetabulum rim and superior pubic ramus (Fig 5).

In the case of reduction in $E_{Cort-OP}/E_{Cort-Normal}$, the posterior/anterior wall (Judet and Letournel classification) was the type of acetabular fracture (Fig 4).

### 3.3. Effect of reduction in the mechanical properties of total bone on the acetabular bone failure

The compressive trabecular bone failure happened in the anterior and posterior acetabular walls in a moderate reduction in $E_{Total-OP}/E_{Total-Normal}$ ($E_{Trab-OP}/E_{Trab-Normal}$: 0.670–0.614 and $E_{Cort-OP}/E_{Cort-Normal}$: 0.835–0.802) (Fig 6). A more reduction in $E_{Total-OP}/E_{Total-Normal}$ ($E_{Trab-OP}/E_{Trab-Normal}$: 0.614–0.472 and $E_{Cort-OP}/E_{Cort-Normal}$: 0.802–0.736) bone failure was also observed in the inferior pubic ramus, and a further reduction ($E_{Trab-OP}/E_{Trab-Normal}$: 0.472–0.340 and $E_{Cort-OP}/E_{Cort-Normal}$: 0.736–0.670) led to compressive bone failure at the ischiopubic ramus (Fig 6). Although tensile trabecular bone failure occurred only in minuscule regions compared with compressive failure, it was seen at the cotyloid fossa, inferior pubic, and ischiopubic ramus with a moderate reduction in $E_{Total-OP}/E_{Total-Normal}$ ($E_{Trab-OP}/E_{Trab-Normal}$: 0.670–0.472 and $E_{Cort-OP}/E_{Cort-Normal}$: 0.835–0.736) (Fig 6). At the extreme reduction in $E_{Total-OP}/E_{Total-Normal}$ ($E_{Trab-OP}/E_{Trab-Normal}$: 0.406–0.304 and $E_{Cort-OP}/E_{Cort-Normal}$: 0.703–0.670), the superior pubic ramus also experienced tensile bone failure (Fig 6). Cortical bone failure (compressive and tensile) occurred at an extreme reduction in $E_{Total-OP}/E_{Total-Normal}$ at the inferior and ischiopubic ramus (Fig 7).

The general types of acetabular fracture at a moderate and high reduction in $E_{Total-OP}/E_{Total-Normal}$ were anterior/posterior wall and anterior with posterior hemi transverse, respectively (Figs 6 and 7).

### 3.4. Effect of reduction in the mechanical properties of bone on the load-transferring mechanism

Results showed that reduction in $E_{Trab-OP}/E_{Trab-Normal}$ had a negligible effect on the maximum transmitted force within the hip and sacroiliac joints as well as within the pubic symphysis (Fig 8A, 8B and 8C).

On the other hand, reduction in $E_{Cort-OP}/E_{Cort-Normal}$ exhibited a notable role in the load-transferring mechanism within all studied joints (Fig 8A, 8B, 8C). For instance, by reducing

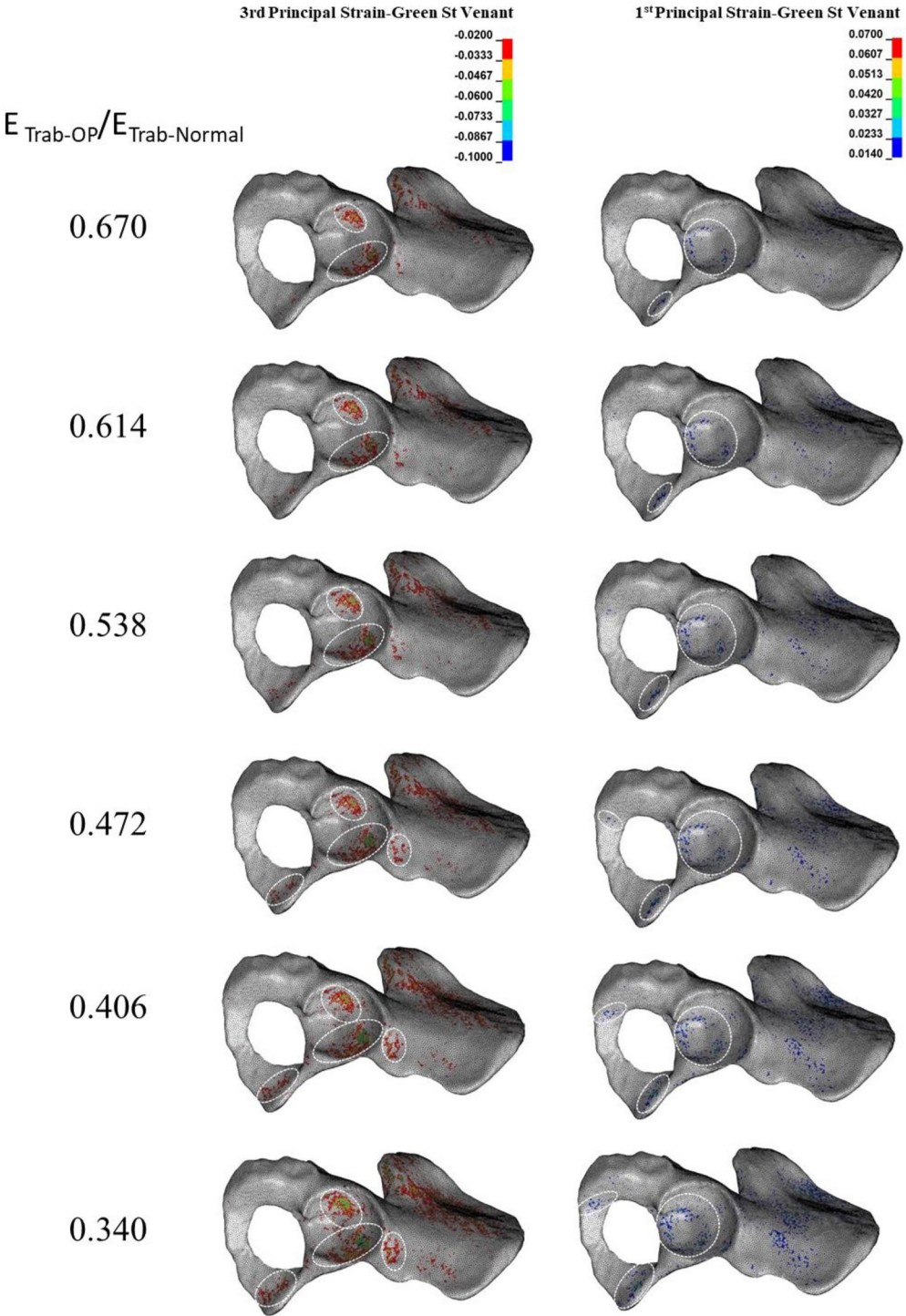

**Fig 2. Effect of reduction in trabecular elastic modulus ($E_{Trab}$) on the compressive (left column) and tensile (right column) trabecular bone failure**. Dashed circles: anatomical landmarks with a high risk of bone failure.

$E_{Cort-OP}/E_{Cort-Normal}$ from 0.835 to 0.670, the maximum transmitted force decreased by 18.6% and 23.9% within the hip and sacroiliac joints, respectively (Fig 8A and 8B). However, the maximum force transfer trend in the sacroiliac joint was not so evident in all levels of $E_{Cort-OP}/$

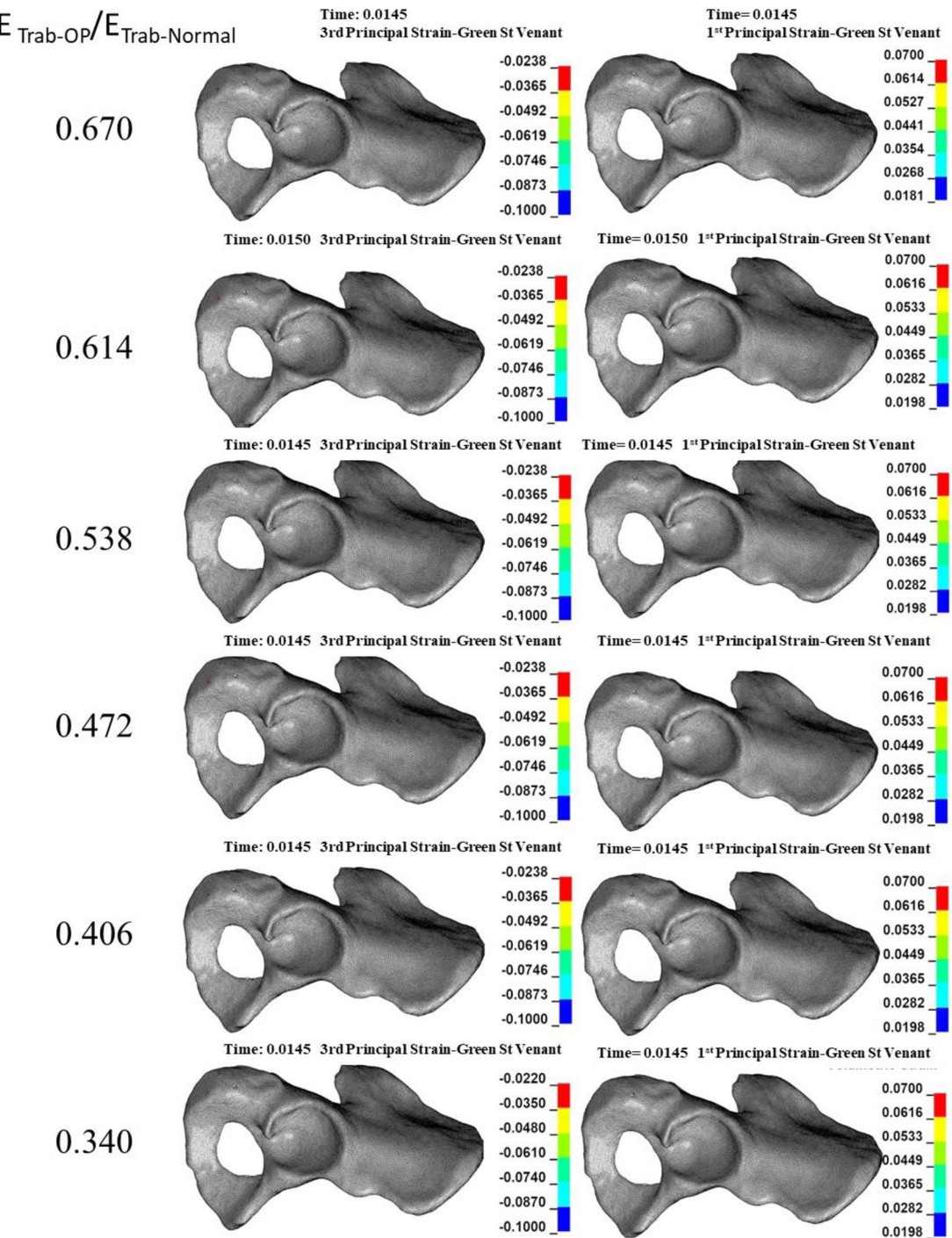

**Fig 3. Effect of reduction in trabecular elastic modulus ($E_{Trab}$) on the compressive (left column) and tensile (right column) cortical bone failure.** Neither tensile nor compressive cortical bone failure was observed.

$E_{Cort-Normal}$ reduction (Fig 8B). In contrast, the maximum transmitted force within the pubic symphysis increased by 13.2% when $E_{Cort-OP}/E_{Cort-Normal}$ reduced from 0.835 to 0.670 (Fig 8C).

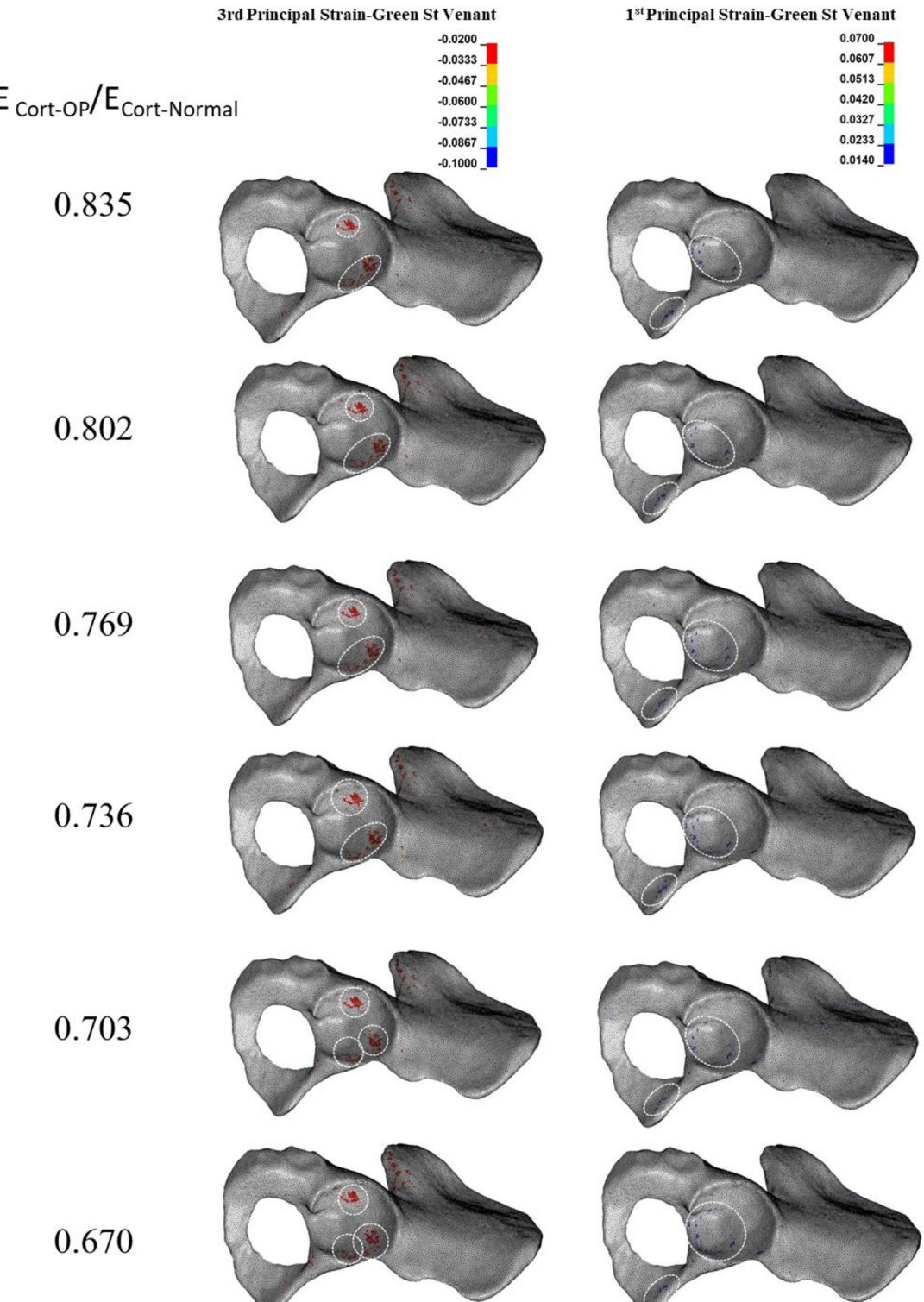

**Fig 4. Effect of reduction in cortical elastic modulus due to osteoporosis (Ecort) on the compressive (left column) and tensile (right column) trabecular bone failure.** Dashed circles: anatomical landmarks with a high risk of bone failure.

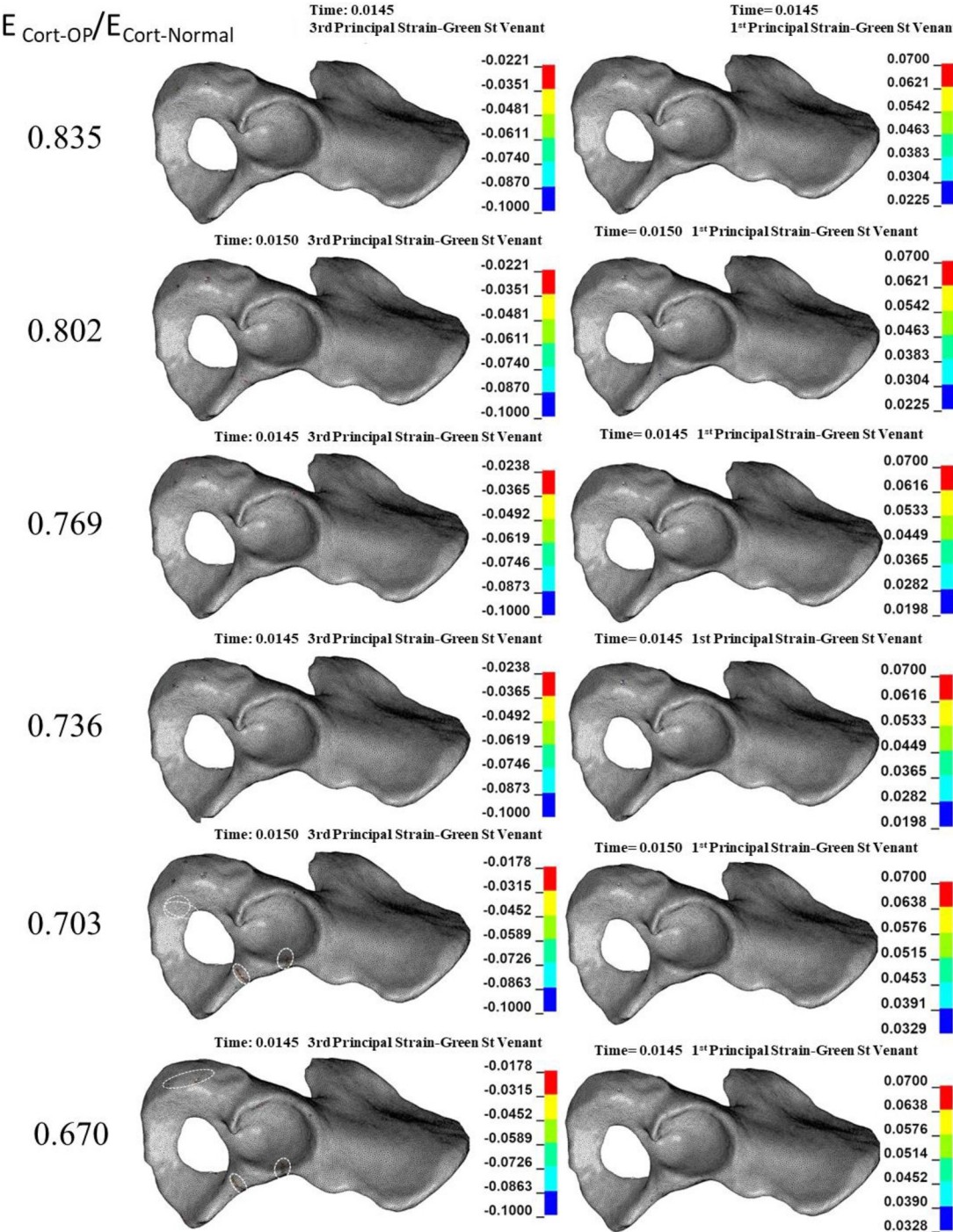

**Fig 5. Effect of reduction in cortical elastic modulus ($E_{cort}$) on the compressive (left column) and tensile (right column) cortical bone failure.** The tensile and compressive strain concentrations were only minor. Dashed circles: anatomical landmarks with a high risk of bone failure.

The effect of reduction in $E_{Total-OP}/E_{Total-Normal}$ on the load-transferring mechanism was similar to that of $E_{Cort-OP}/E_{Cort-Normal}$. Within the hip and sacroiliac joints, the maximum transmitted force was lowered by reducing $E_{Total-OP}/E_{Total-Normal}$ (Fig 8A and 8B). In the case

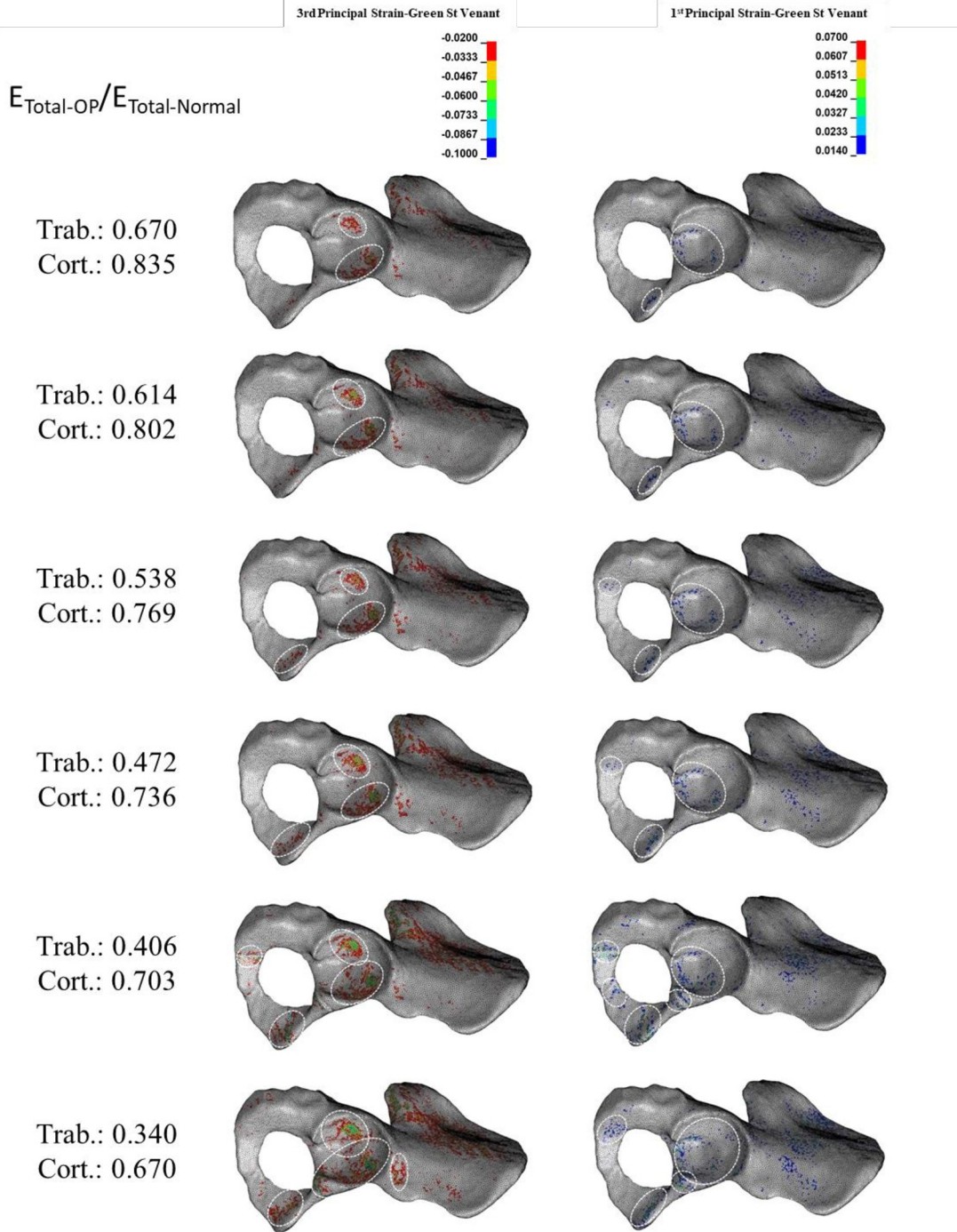

**Fig 6. Effect of reduction in total (Trab. + Cort.) elastic modulus ($E_{Total}$) on the compressive (left column) and tensile (right column) trabecular bone failure.** Dashed circles: anatomical landmarks with a high risk of bone failure.

of the pubic symphysis, reduction in $E_{Total-OP}/E_{Total-Normal}$, especially at high severity levels, led to an increase in transmitted force in this joint (Fig 8C).

The time history plots of the transmitted load are presented in Appendix B: Fig B1-3 in S1 File, to provide insights into the load transferring mechanism within the hip, sacroiliac, and pubic symphysis joints.

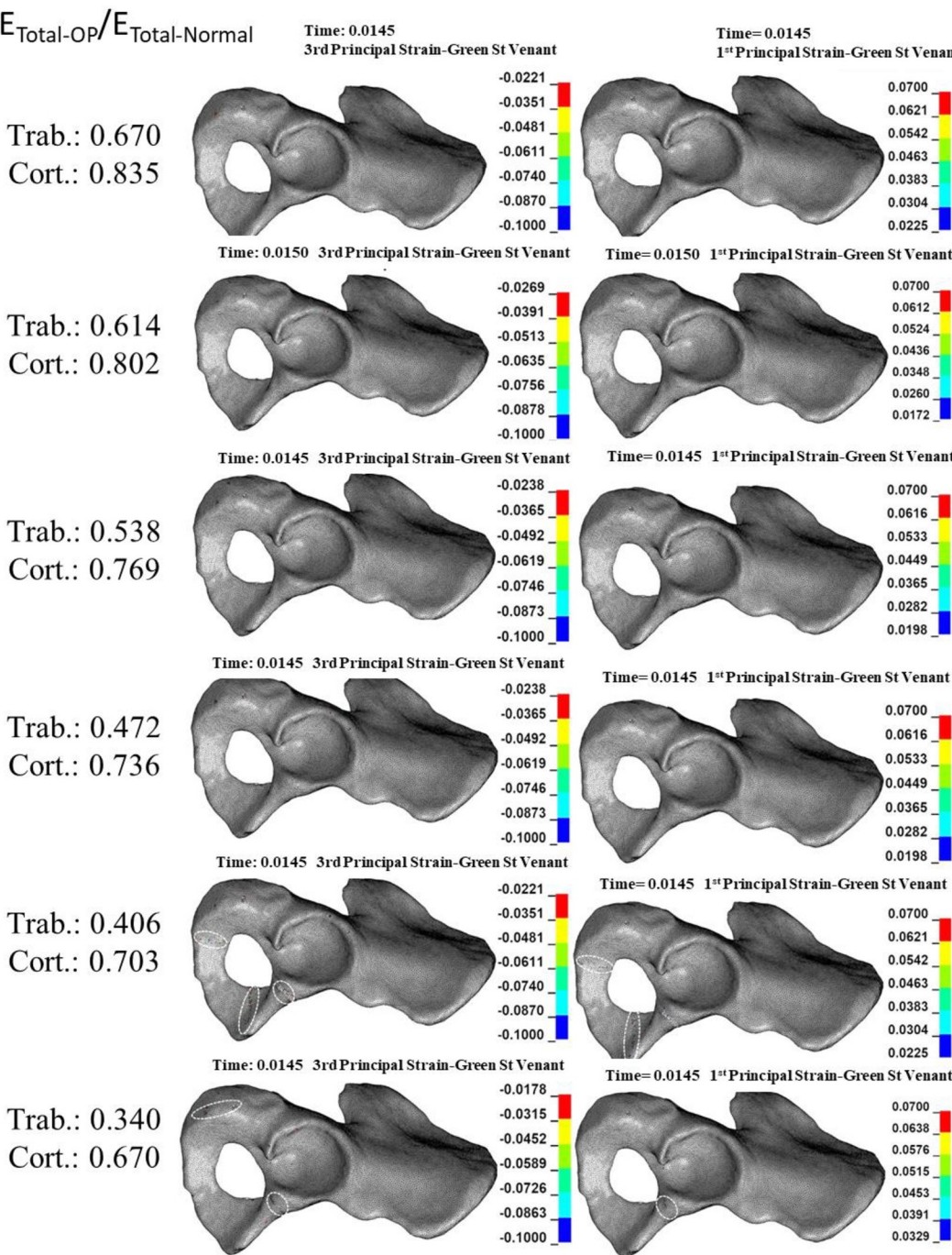

**Fig 7. Effect of reduction in total (Trab. + Cort.) elastic modulus ($E_{Total}$) on the compressive (left column) and tensile (right column) cortical bone failure.** Dashed circles: anatomical landmarks with a high risk of bone failure.

## 4. Discussion

This study, through a parametric finite element approach, investigated the effect of OP-related reduction in the mechanical properties of bone on the bone failure intensity and pattern (fracture type) and load-transferring mechanism within the pelvic girdle joints. To the best of our

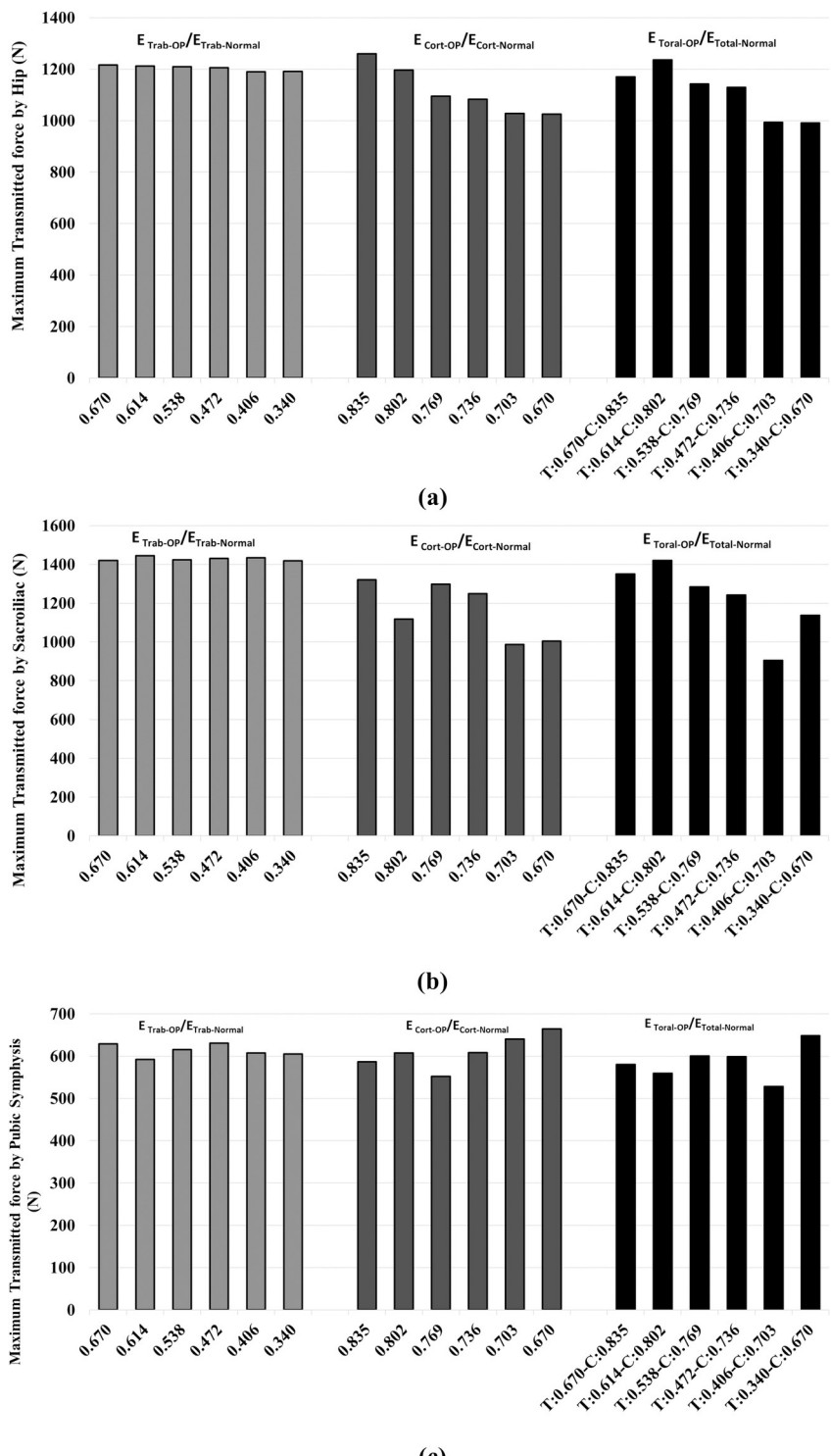

**Fig 8.** Effect of reduction in trabecular ($E_{Trab}$), cortical ($E_{cort}$), and total ($E_{Total}$) elastic on the transmitted load (contact force) by a) hip, b) sacroiliac, and c) pubic symphysis.

knowledge, this is the first study that evaluated the effect of reduction in mechanical properties of bone caused by OP on low-energy acetabular fractures.

In the absence of cadaveric experiments such as those conducted by Fleps et al. [11] or Askarnejad et al. [48], the model was validated against the literature [11, 12, 28] and presented in Appendix C, Fig C1, and Table C1 in S1 File. Also, our previous study [16] showed the ability of the adopted model to predict and explain clinical and experimental data.

The results showed that whereas a reduction in the mechanical properties of trabecular bone significantly affects only trabecular bone failure (Figs 2 and 3), reduction in the mechanical properties of cortical bone influences both trabecular (Fig 4) and cortical (Fig 5) bone failures. It suggests that similar to the femoral neck [49], cortical bone has a critical contribution to the prevention of acetabular fracture among elderly people. According to Rinne et al. [50], the incidence of low-energy acetabular fracture is much higher for people aged 65 or more, which is the age range reported for cortical bone failure [17]. According to our results, acetabular bone failure is substantially affected by the reduction in the mechanical properties of both trabecular and cortical bones, which happens in real life (Figs 6 and 7). However, reduction in mechanical properties of trabecular bone affected bone failure only at the trabecular level (Fig 3); the role of the trabecular bone in withstanding acetabular fractures should not be underestimated. Our results showed that in all cases (reduction in the mechanical properties of trabecular, cortical, and total bone), trabecular bone failure occurred at the cotyloid fossa, which has been shown as a region with extremely high-stress concentration [51]. Also, Dalstra et al. [52] have shown that the anatomical distribution of trabecular bone density at the sacroiliac joint and pubic symphysis areas is greater than in other regions and enables resistance to high local stresses. This may explain why compressive trabecular bone failure at the superior rim of the acetabulum and superior/inferior pubic ramus can be observed only at a high reduction in the trabecular elastic modulus when trabecular bone density is decreased notably (Figs 2 and 6). In the current study, cortical bone failure was observed only in the case of a high reduction in bone mechanical properties (Figs 5 and 7), while trabecular bone failure (with different severity and failure location) occurred in all reductions in the mechanical properties at the acetabulum. Therefore, it seems that at the moderate reduction in the elastic modulus, acetabular fractures initiate at the trabecular bone level, and then, by more severe reduction in the mechanical properties, in addition to the trabecular, bone failure expands and engages the cortical bone, similar to the mechanism of the proximal femur during a low-energy sideways fall [11].

The risk of acetabular fracture [9, 50] and the prevalence of osteoporotic fracture [53] both increase considerably with age. According to our findings, the type of acetabular fracture changes by increasing reduction in the mechanical properties of bone. At lower stages of OP-related reduction in the mechanical properties of bone, we observed posterior/anterior wall fractures, which account for 13% of all acetabular fractures among elderly people [9]. At intensive reductions in the mechanical properties of bone (older patients), the bone failure pattern resembles anterior with posterior hemi transverse and both-column acetabular fracture types. Firoozabadi et al. [9] identified these types of acetabular fractures (anterior with posterior hemi transverse = 35% and both columns = 34%) as the most prevalent, consistent with the current study findings.

Finally, this study suggests that the effect of reduction in the mechanical properties of cortical bone on the load-transferring mechanism within the pelvic joints is more critical than that of trabecular bone, and by increasing reduction of mechanical properties of cortical bone, bigger shares of impact load are transmitted via the pubic symphysis and smaller shares via the sacroiliac joint. Dalstra and Huiskes [51] stated that the cortical bone handles a major part of the transmitted load in the pelvis, confirming our finding.

This study has several limitations. First, although including muscle forces could increase the model accuracy, most elderly people are subjected to a free-fall and are not able to use their muscles to control or block the fall. Considering this fact and owing to the absence of clinical data regarding the muscle force magnitude and its recruitment pattern during a low-energy sideways fall, muscle forces were not included. Second, although bone anisotropy and cortical bone heterogeneity due to the complex shape of the pelvic bone were not included directly, the used element-by-element material mapping techniques for trabecular bone, based on HU, makes it structurally anisotropic. Third, to decrease the associated computational costs, only half of the soft tissue was modeled. Also, owing to the use of abdominal CT images, the upper and lower extremities stiffness and weight (inertia) were included by using an effective simplifying method (mass-spring-dashpot system) instead. Fourth, although patient-specific models could yield more accurate results, owing to the aim (relative effect of OP-related reduction in mechanical properties of bone on the acetabular bone failure) and type (parametric) of the current study, only a median model was developed. Fifth, while using more advanced criteria such as the coupled criterion method (strain and energy) may improve bone failure prediction, owing to the lack of data on the energy-based bone failure criterion in this study, the critical strain failure criterion was considered for assessing bone failure. Moreover, previous studies such as Fleps et al. [11] successfully used the strain failure criterion in the prediction of proximal bone failure. Sixth, although OP substantially alters the bone architecture (thickness of trabecula and the connectivity level of the trabecular network), from a macrolevel perspective, the reduction in the bone apparent density may be considered as the effect of change in the bone architecture. In addition, owing to the lack of comprehensive data regarding changes in the pelvic (acetabulum) architecture by the increasing bone loss, only changes in apparent densities and other mechanical properties of bone were considered. Finally, since the aim of this study was to conduct a parametric (relative) study of the effect of reduction in the apparent density and other mechanical properties of bone on low-energy acetabular fractures and not to precisely predict bone fractures, the model was validated against the literature and was not directly validated because of the lack of cadaveric tests.

In conclusion, this study increases our understanding of the effect of reduction in the mechanical properties of bone due to OP progression on low-energy acetabular fractures. According to the current study, reduction in the mechanical properties of cortical bone (and consequently other mechanical properties) affects both the cortical and trabecular bones, and in the case of severe reduction, the acetabular fracture type could be highly complicated, which needs extensive surgical intervention and rehabilitation period. Finally, it seems an osteoporotic hemipelvis at the impact side decreases the transmitted load to the sacrum and might lower the risk of sacral fractures, which would be of our future research interests.

## Supporting information

**S1 File.**
(DOCX)

## Acknowledgments

CSC–IT Center for Science LTD, Finland, is acknowledged for providing FE software and computational resources.

## Author Contributions

**Conceptualization:** Shahab Khakpour, Mika E. Mononen, Rami K. Korhonen, Timo Jämsä.

**Data curation:** Shahab Khakpour, Timo Jämsä.

**Formal analysis:** Shahab Khakpour.

**Funding acquisition:** Rami K. Korhonen, Timo Jämsä.

**Investigation:** Shahab Khakpour.

**Methodology:** Shahab Khakpour, Amir Esrafilian, Timo Jämsä.

**Project administration:** Timo Jämsä.

**Resources:** Timo Jämsä.

**Software:** Shahab Khakpour, Petri Tanska, Mika E. Mononen.

**Supervision:** Rami K. Korhonen, Timo Jämsä.

**Validation:** Shahab Khakpour, Timo Jämsä.

**Visualization:** Shahab Khakpour, Mika E. Mononen.

**Writing – original draft:** Shahab Khakpour.

**Writing – review & editing:** Shahab Khakpour, Amir Esrafilian, Petri Tanska, Mika E. Mononen, Rami K. Korhonen, Timo Jämsä.

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
