## [Decision Letter · Decision Letter 0]

20 Dec 2021

PONE-D-21-33350Effect of osteoporosis-related reduction in the mechanical properties of bone on the acetabular fracture during a sideways fall: A parametric finite element approachPLOS ONE

Dear Dr. Shahab Khakpour,

Thank you for submitting your manuscript to PLOS ONE. After careful consideration, we feel that it has merit but does not fully meet PLOS ONE’s publication criteria as it currently stands. Therefore, we invite you to submit a revised version of the manuscript that addresses the points raised during the review process.

Please submit your revised manuscript by Feb 03 2022 11:59PM. If you will need more time than this to complete your revisions, please reply to this message or contact the journal office at plosone@plos.org. Please include the following items when submitting your revised manuscript:A rebuttal letter that responds to each point raised by the academic editor and reviewer(s). You should upload this letter as a separate file labeled 'Response to Reviewers'.A marked-up copy of your manuscript that highlights changes made to the original version. You should upload this as a separate file labeled 'Revised Manuscript with Track Changes'.An unmarked version of your revised paper without tracked changes. You should upload this as a separate file labeled 'Manuscript'.If applicable, we recommend that you deposit your laboratory protocols in protocols.io to enhance the reproducibility of your results. Protocols.io assigns your protocol its own identifier (DOI) so that it can be cited independently in the future. For instructions see: https://journals.plos.org/plosone/s/submission-guidelines#loc-laboratory-protocols. Additionally, PLOS ONE offers an option for publishing peer-reviewed Lab Protocol articles, which describe protocols hosted on protocols.io. Read more information on sharing protocols at https://plos.org/protocols?utm_medium=editorial-email&utm_source=authorletters&utm_campaign=protocols.

We look forward to receiving your revised manuscript.

Kind regards,

Ewa Tomaszewska, DVM Ph.D

Academic Editor

PLOS ONE

“This project was supported by the I4Future doctoral program under the Marie Skłodowska-Curie grant agreement No 713606.”

We note that you have provided information within the Acknowledgements Section. Please note that funding information should not appear in the Acknowledgments section or other areas of your manuscript. We will only publish funding information present in the Funding Statement section of the online submission form.

 “Shahab Khakpour:

This project received funding from European Union's Horizon 2020 research and innovation programme under the Marie Skoldowska-Curie grant agreement No 713606.

https://www.oulu.fi/i4future/

4. PLOS requires an ORCID iD for the corresponding author in Editorial Manager on papers submitted after December 6th, 2016. Please ensure that you have an ORCID iD and that it is validated in Editorial Manager. To do this, go to ‘Update my Information’ (in the upper left-hand corner of the main menu), and click on the Fetch/Validate link next to the ORCID field. This will take you to the ORCID site and allow you to create a new iD or authenticate a pre-existing iD in Editorial Manager. Please see the following video for instructions on linking an ORCID iD to your Editorial Manager account: https://www.youtube.com/watch?v=_xcclfuvtxQ.

Reviewers' comments:

Reviewer's Responses to Questions

**Comments to the Author**

1. Is the manuscript technically sound, and do the data support the conclusions?

Reviewer #1: Yes

Reviewer #2: Yes

2. Has the statistical analysis been performed appropriately and rigorously? 

Reviewer #1: Yes

Reviewer #2: Yes

3. Have the authors made all data underlying the findings in their manuscript fully available?

Reviewer #1: No

Reviewer #2: Yes

4. Is the manuscript presented in an intelligible fashion and written in standard English?

Reviewer #1: Yes

Reviewer #2: Yes

5. Review Comments to the Author

Reviewer #1: The article entitled „Effect of osteoporosis-related reduction in the mechanical properties of bone on the acetabular fracture during a sideways fall: A parametric finite element approach” is an interesting work on the finite element approach to the analysis of reduction in mechanical properties of bones and its influence on fracture.

The Authors introduce the problem based on many excellent works by other researchers and their research. Present all the assumptions of the experiment concerning literature and support their attitudes. The work is well planned, and the experiment is carried out correctly. The quality of the description and presentation is on a very high level.

I strongly recommend the work for publication by PlosOne.

Reviewer #2: The text is well written, clear and easy to comprehend and follow despite its length. I am enthusiastic about works like this and strongly recommend the publication of this work.

Just some minor comments:

1. Consider unifying the nomenclature, e.g.:

- lowercase epsilon (table) or lunatic epsilon (Enns-Bray formulas) in App A.

- avoid to mix different methods of presenting numerical values – L227-230 SI prefixes (ms) or decimal notation (10^(-8))

- E_Total-OP/E_Normal or E_Total-OP/E_Total_Normal throughout the main text and App A.

2. L218 Correct to “Green-St Venant strain tensor”

3. Figs B1-B3 in App - Some decimal points in figures of total elastic modulus legend (on the right) are missing.

4. references are not formatted in accordance with journal requirements

5. Ref [15] – Journal title is missing

6. Figures 2-7 What do the circles mean ? Some explanation in figure’s footnote will be helpful.

7. Fig 3 and 6 – As no points can be seen on the most of the figures, also explanation in figure’s footnote will be helpful.

6. PLOS authors have the option to publish the peer review history of their article (what does this mean?). If published, this will include your full peer review and any attached files.

Reviewer #1: No

Reviewer #2: No

---

## [Author Response · Author response to Decision Letter 0]

17 Jan 2022

Dear Professor Tomaszewska, 

Thank you for the opportunity to revise our manuscript entitled “Effect of osteoporosis-related reduction in the mechanical properties of bone on the acetabular fracture during a sideways fall: A parametric finite element approach”. We appreciate the careful review and constructive suggestions. The manuscript has certainly benefited from the insightful revision suggestions. Based on the comments from the referees, we have made changes to the manuscript, which are detailed below. We are confident that we have addressed all the comments carefully and provided clear responses. 

Following this letter are the reviewers’ comments (in bold) with dedicated responses for each comment, including how and where the text is modified. The changes made to the manuscript are reported in italics in the response letter and marked with the Track Changes feature of the Microsoft office in the revised manuscript. 

Your time and considerations are greatly appreciated. 

Yours Sincerely, 

Shahab Khakpour, on behalf of all the co-authors 

Research Unit of Medical Imaging, Physics, and Technology (MIPT) 

Faculty of Medicine, University of Oulu, Oulu, Finland

 

Reviewer #2: 

General Comment

The text is well written, clear and easy to comprehend and follow despite its length. I am enthusiastic about works like this and strongly recommend the publication of this work. Just some minor comments. 

Response: 

We do appreciate your constructive and positive comments. We made our best to completely address your comments.

1 Consider unifying the nomenclature, e.g.:

- lowercase epsilon (table) or lunatic epsilon (Enns-Bray formulas) in App A.

- avoid to mix different methods of presenting numerical values 

– L227-230 SI prefixes (ms) or decimal notation(10^(-8))

- E_Total-OP/E_Normal or E_Total-OP/E_Total_Normal throughout the main text and App A.

Response: 

Thank you for the comment. We unified the nomenclature and notations throughout the manuscript. 

Changes: 

- All epsilons in Appendix A were unified to the lowercase epsilon. 

- “s” was changed to “ms” in line 230 as follows:

“3.36×10-5 ms in the simulations”

Also, the units of horizontal axes in Fig B 1-3 were changed from s to ms. 

- All E_Total-OP/E_Normal were changed to E_Total-OP/E_Total_Normal throughout the text, Fig 6-8, and Fig B 1-3 (appendix B). 

2 L218 Correct to “Green-St. Venant strain tensor”

Response: 

Thank you for the comment. 

Changes: 

The typo was corrected accordingly in Line 219 as follows:

“element softening (first principal strains of the Green-St.Vernant strain tensor higher than 0.014”

3 Figs B1-B3 in App - Some decimal points in figures of total elastic modulus legend (on the right) are missing

Response: 

Thank you for the comment and for mentioning the needed change. 

Changes: 

The legends of Fig B1-B3 were crosschecked and corrected accordingly.

4 References are not formatted in accordance with journal requirements

Response: 

Thank you for the comment.

Changes: 

References were formatted according to the journal requirements (Vancouver style). 

5 Ref [15] – Journal title is missing

Response: 

We appreciate your comments. 

Changes: 

The reference was corrected. 

6 Figures 2-7 What do the circles mean ? Some explanation in figure’s footnote will be helpful.

Response: 

Thank you for the comment. Adding an explanation in the figure’s footnote would enhance the readability of the figures. 

Changes:

Explaining statements were added to Figure 2-7 as follows:

“Dashed circles: anatomical landmarks with a high risk of bone failure” . 

7 Fig 3 and 6 – As no points can be seen on the most of the figures, also explanation in figure’s footnote will be helpful.

Response: 

Thank you for the comment. As mentioned by you, adding the figure’s footnote would help the reader to understand them better. We also think that you addressed Fig 5, instead of Fig 6. 

Changes:

Additional explanation was added to Fig 3 and 5 as follows:

Fig 3: “Neither tensile nor compressive cortical bone failure was observed.”

Fig 5: “The tensile and compressive strain concentrations were only minor.”

---

## [Decision Letter · Decision Letter 1]

20 Jan 2022

Effect of osteoporosis-related reduction in the mechanical properties of bone on the acetabular fracture during a sideways fall: A parametric finite element approach

PONE-D-21-33350R1

Dear Dr. Shahab Khakpour,

We’re pleased to inform you that your manuscript has been judged scientifically suitable for publication and will be formally accepted for publication once it meets all outstanding technical requirements.

Kind regards,

Ewa Tomaszewska, DVM Ph.D

Academic Editor

PLOS ONE

Additional Editor Comments (optional):

Reviewers' comments:

Reviewer's Responses to Questions

**Comments to the Author**

1. If the authors have adequately addressed your comments raised in a previous round of review and you feel that this manuscript is now acceptable for publication, you may indicate that here to bypass the “Comments to the Author” section, enter your conflict of interest statement in the “Confidential to Editor” section, and submit your "Accept" recommendation.

Reviewer #2: (No Response)

2. Is the manuscript technically sound, and do the data support the conclusions?

Reviewer #2: Yes

3. Has the statistical analysis been performed appropriately and rigorously? 

Reviewer #2: Yes

4. Have the authors made all data underlying the findings in their manuscript fully available?

Reviewer #2: Yes

5. Is the manuscript presented in an intelligible fashion and written in standard English?

Reviewer #2: Yes

6. Review Comments to the Author

Reviewer #2: I would like to thank the Authors for reviewing and accepting all the comments and suggestions. In my opinion the article is now acceptable for publication.

7. PLOS authors have the option to publish the peer review history of their article (what does this mean?). If published, this will include your full peer review and any attached files.

Reviewer #2: No

---

## [Editor Report · Acceptance letter]

26 Jan 2022

PONE-D-21-33350R1 

Effect of osteoporosis-related reduction in the mechanical properties of bone on the acetabular fracture during a sideways fall: A parametric finite element approach 

Dear Dr. Khakpour:

I'm pleased to inform you that your manuscript has been deemed suitable for publication in PLOS ONE. Congratulations! Your manuscript is now with our production department. 

Kind regards, 

on behalf of

Professor Ewa Tomaszewska 

Academic Editor

PLOS ONE